# Nationwide Trends in Demographics, Comorbidities, and Mortality Among Elderly Patients with Heart Failure with Preserved Ejection Fraction Hospitalized with Cardiac Arrest

**DOI:** 10.3390/jpm15110559

**Published:** 2025-11-18

**Authors:** Adil Sarvar Mohammed, Maya Asami Takagi, Umera Yasmeen, Aashna Gandhi, Ayesha Firdous Shafiulla Khan, Apurva Popat, Rupak Desai, Shrinivas Kambali, Ahmad Khalil A. Koshak, Sohaib Mandoorah

**Affiliations:** 1Department of Internal Medicine, Central Michigan University, 1000 Houghton Ave, Saginaw, MI 48602, USA; 2Department of Internal Medicine, University of California (Davis), 4150 V Street, Sacramento, CA 95817, USA; 3Department of Medicine, Mamata Medical College, Rotary Nagar, Khammam 507002, India; umera.uy@gmail.com; 4GCS Medical College, Hospital & Research Centre, Opposite Divisional Railway Manager’s Office, NR Railway Station, Ahmedabad 380025, India; draashnagandhi2000@gmail.com; 5Department of Internal Medicine, University of Tennessee Health Science Center, 956 Court Ave, Memphis, TN 38163, USA; 6Department of Cardiology, Marshfield Clinic Health System, 1000 N Oak Ave, Marshfield, WI 54449, USA; popat.apurva@marshfieldclinic.org; 7Independent Researcher, Atlanta, GA 30033, USA; 8Department of Pulmonary and Critical Care Medicine, MyMichigan Medical Center, 1000 Houghton Ave, Saginaw, MI 48601, USA; 9Department of Internal Medicine, University of Tabuk, Prince Fahd Bin Sultan Rd, Tabuk 47512, Saudi Arabia; akoshak@ut.edu.sa; 10International Medical Center, JBRB4238, 4238 Nour Alhuda, Jeddah 23214, Saudi Arabia; 11Department of Critical Care Medicine, MyMichigan Medical Center, 1000 Houghton Ave, Saginaw, MI 48601, USA; 12Department of Critical Care Medicine, Central Michigan University, 1000 Houghton Ave, Saginaw, MI 48602, USA

**Keywords:** heart failure with preserved ejection fraction, cardiac arrest, mortality, hospitalization, disparities, elderly

## Abstract

**Background:** Heart failure with preserved ejection fraction (HFpEF) is a major cause of hospitalization and mortality in older adults. Sudden cardiac arrest (SCA) is a leading cause of death in this population, yet national trends in incidence, outcomes, and disparities remain poorly defined. **Methods:** We performed a retrospective cohort study using the National Inpatient Sample from 2016 to 2020. Hospitalizations for patients aged ≥65 years with HFpEF and in-hospital cardiac arrest (CA) were identified using ICD-10-CM codes. Demographics, comorbidities, hospital outcomes, and temporal trends were examined. The primary outcome was in-hospital mortality. Secondary outcomes included length of stay, hospital charges, and discharge disposition. **Results:** Among 7,738,108 HFpEF admissions, 93,440 (1.2%) involved CA. Incidence rose from 1.1% in 2016 to 1.5% in 2020 (36% relative increase). The median age was 81 years; 54% were female, 70% White, 19% Black, and 8% Hispanic. CA incidence increased across all groups, with the largest relative rises among Native American (1.0% to 1.9%), Black (1.7% to 2.3%), and Hispanic patients (1.4% to 2.0%). In-hospital mortality was high, increasing from 58.2% to 61.7% over the study period (*p* < 0.001). Mortality rose most steeply among Black and low-income patients. Comorbidity patterns shifted toward greater metabolic complexity, including higher rates of complicated diabetes, hypertension, hyperlipidemia, and obesity. **Conclusions:** Elderly patients hospitalized with HFpEF are experiencing rising rates of in-hospital CA and persistently high mortality, with marked racial and socioeconomic disparities. These findings highlight the need for better risk stratification, targeted metabolic and inflammatory therapies, and more equitable care delivery.

## 1. Introduction

Heart failure (HF) is a leading cause of hospitalization and mortality in individuals over 65 years, with nearly half of affected patients exhibiting a preserved ejection fraction (HFpEF). Despite its growing prevalence, particularly among older adults with multiple co-morbidities, HFpEF remains a complex and heterogenous syndrome with limited therapeutic options and a prognosis comparable to that of HF with reduced ejection fraction (HFrEF) [1,2,3,4,5,6,7].

HFpEF is recognized as a distinct syndrome involving systemic inflammation, endothelial dysfunction, and increased ventricular and arterial stiffness, which together lead to impaired left ventricular (LV) relaxation and elevated filling pressures despite a preserved ejection fraction [5,8]. Obesity, particularly central obesity, plays a central role by promoting myocardial fibrosis, impairing ventricular relaxation, and contributing to right ventricular dysfunction [5,9,10,11,12,13]. These abnormalities are compounded by diminished physiologic reserve in the elderly, including impaired skeletal muscle oxygen extraction and vascular compliance, which explain the hallmark symptoms of external dyspnea and fatigue [13,14,15,16,17,18].

Given the high risk of in-hospital cardiac arrest (CA) and mortality among elderly HFpEF patients, improved risk stratification is urgently needed. Potential predictors of CA include conduction abnormalities (i.e., third-degree AV block, left bundle branch block), comorbid liver disease, obesity, and metabolic syndrome. While the current study primarily describes population-level trends, these data may inform future risk models for early identification of high-risk HFpEF patients.

The hospitalization burden associated with HFpEF continues to rise, particularly among older adults. Data from the Nationwide Inpatient Sample (NIS) between 2003 and 2012 show that HFpEF accounted for over half of acute HF hospitalizations in patients over the age of 75 years [19]. More recent analyses of inpatient trends in the United States reveal an increase in HFpEF-related admissions, rising from approximately 189,000 in 2008 to 495,000 in 2018, highlighting its impact on hospitalization burden [20].

Beyond readmissions and the patients’ quality of life, sudden cardiac arrest (SCA) has emerged as a leading cause of death in patients with HFpEF, responsible for about 25% of overall mortality in this population. In-hospital data show that 1.48% of HFpEF admissions are complicated by SCA, which is associated with significantly higher in-hospital mortality (25.9% vs. 1.6%). Notably, key predictors of in-hospital cardiac arrest (CA) include third-degree atrioventricular block, left bundle branch block, and comorbid liver disease [21]. Compared to those with HFrEF, patients with HFpEF are more likely to present with non-shockable rhythms, such as pulseless electrical activity or asystole, which further contribute to poor survival outcomes [22].

These trends highlight the need for improved risk stratification and tailored in-hospital management strategies for elderly patients with HFpEF, particularly in the context of acute decompensation and CA. Although rhythm-specific data are not available in the NIS, identification of demographic and comorbidity patterns may support targeted monitoring and interventions.

## 2. Materials and Methods

We conducted a retrospective cohort study using data from the NIS between 2016 and 2020. To address potential COVID-19-related biases, sensitivity analyses excluding 2020 (pre-pandemic 2016–2019) were also performed. The NIS, developed by the Agency for Healthcare Research and Quality as part of the Healthcare Cost and Utilization Project, is the largest publicly available all-payer inpatient database in the United States. It captures data from approximately 7 million hospitalizations annually, representing a 20% stratified sample of discharges from community hospitals across the country, excluding rehabilitation and long-term acute care facilities, and approximates 97% of the United States population. The dataset includes a broad range of clinical and nonclinical information, including ICD-10-CM diagnosis and procedure codes, patient demographics, hospital characteristics, insurance type, discharge disposition, hospital charges, and comorbidity and severity measures. We used the Clinical Classification Software Refined (CCSR) v2025.1 tool to categorize ICD-10-CM codes into clinically relevant groupings. Because the dataset is fully de-identified, institutional review board approval was not required.

We included hospitalizations for patients aged 65 years and older with a concurrent diagnosis of HFpEF and CA, identified using ICD-10-CM codes. We extracted demographic variables including age, sex, race, and median household income quartile by ZIP code, as well as hospital region and insurance status. We also identified comorbidities such as diabetes (with and without complications), hypertension (complicated and uncomplicated), prior myocardial infarction (MI), stroke, hyperlipidemia, obesity, peripheral vascular disease, chronic kidney disease, hypothyroidism, malignancy, and substance use disorders.

The primary outcome was all-cause in-hospital mortality. Secondary outcomes included hospital length of stay (LOS), total hospital charges, and discharge disposition (i.e., discharge to a skilled nursing facility). Baseline characteristics were compared between patients with and without in-hospital CA, with results presented in Appendix A.

We used descriptive statistics to summarize patient characteristics and clinical variables by year and applied linear-by-linear association testing to assess trends over time. All analyses accounted for the NIS’s complex survey design, and statistical significance was defined as a two-tailed *p*-value < 0.05.

## 3. Results

A total of 7,738,108 hospital admissions for HFpEF were identified from 2016 to 2020. Among these, 93,440 patients (1.2%) experienced CA in the hospital, while the remaining 7,644,668 (98.8%) did not. From 2016 to 2020, the number of HFpEF admissions increased annually, peaking in 2019, before declining slightly in 2020. The incidence of CA rose from 1.1% in 2016 to 1.5% in 2020 (Figure 1). In a sensitivity analysis excluding 2020 to account for potential pandemic-related effects, similar trends were observed, with the rate of in-hospital cardiac arrest increasing from 1.1% in 2016 to 1.4% in 2019 (compared with 1.5% in 2020 in the full cohort). This represents a relative increase of about 36% in CA incidence over the 5-year period.

Demographic and temporal trends in CA incidence and in-hospital mortality among elderly patients with HFpEF are summarized in Table 1, with additional subgroup analyses presented in Appendix A. The median age was 81 years (IQR 77–84), and the cohort was 54% female. Among in-hospital CA cases, 54% were female and 46% male, as shown in Table 1. Racial distribution was predominantly White (70%), followed by Black (19%) and Hispanic (8%), with smaller proportions identifying as Native American and Asian American/Pacific Islander. A greater proportion of patients were in the lowest income quartile (31%), with most covered by Medicare (90%) and residing in urban areas (93%). Over the five-year period, CA incidence increased significantly across all demographic groups. Among females, incidence rose from 1.0% to 1.3%, and among males from 1.3% to 1.7%. Native American patients experienced the highest relative increase (1.0% to 1.9%), followed by Black (1.7% to 2.3%) and Hispanic patients (1.4% to 2.0%). CA incidence in the lowest income quartile increased from 1.2% to 1.7%, and the second income quartile saw a 50% relative rise (1.0% to 1.5%).

Baseline comparisons between CA and non-CA patients are summarized in Appendix A. Patients experiencing CA were older, had higher rates of complicated diabetes, complicated hypertension, and obesity, and were more likely to be from lower-income quartiles.

In-hospital mortality among HFpEF patients with CA was high and increased overall from 58.2% in 2016 to 61.7% in 2020 (*p* for trend < 0.001). Mortality increased from 58.9% to 62.3% among females and from 57.4% to 61.1% among males. Black patients experienced the largest increase in mortality (56.9% to 62.9%), while White patients had a smaller increase (58.4% to 60.8%). The highest mortality rate in 2020 was observed in Hispanic patients (65.4%), although this trend was not statistically significant. Patients in the lowest income quartile had the largest mortality increase among income groups (57.3% to 63.5%, *p* < 0.001). Notably, rural patients experienced a decline in in-hospital mortality from 57.5% to 33.7% over the same period (*p* = 0.001), a trend that differs from other subgroups and warrants further investigation.

The clinical profile of patients hospitalized with HFpEF and CA evolved significantly over the study period from 2016 to 2020. As summarized in Table 2, there was a notable decline in the prevalence of certain comorbidities, including diabetes without chronic complications (17.8% to 7.3%), uncomplicated hypertension (25.8% to 1.0%), prior myocardial infarction (10.6% to 8.3%), and prior stroke/transient ischemic attack (9.4% to 8.7%) (all *p* < 0.001). Conversely, the burden of metabolic and cardiovascular complexity increased, demonstrated by rising rates of diabetes with chronic complications (28.0% to 42.9%), complicated hypertension (54.4% to 81.8%), hyperlipidemia (46.5% to 51.0%), and obesity (21.8% to 25.2%) over the same period (all *p* < 0.001). Further detailed trends for additional comorbidities and outcomes are provided in Appendix A.

Despite a modest decrease in median hospital length of stay from 7 days in 2016 to 6 days in 2020 (*p* < 0.001), hospital charges increased by approximately 6% during the study period, as illustrated in Table 3. Among survivors, 23.2% were discharged to skilled nursing or intermediate care facilities, reflecting substantial post-acute care needs in this high-risk population.

## 4. Discussion

HFpEF represents a growing contributor to acute care utilization and mortality amongst the elderly population. In this large, nationally representative cohort of elderly patients with HFpEF hospitalized with CA, we found a significantly high in-hospital mortality rate of 58.9%, which increased over the five-year study period. The burden of CA in the HFpEF population parallels broader epidemiological trends illustrating that HFpEF accounts for over half of heart failure admissions in patients aged 75 and older [23]. Our findings emphasize the need for urgent therapeutic attention and better risk stratification in this age group.

### 4.1. Demographics and Clinical Trends

Women made up a higher proportion of in-hospital CA cases compared to men, contrasting prior studies identifying male sex as a predictor of SCD and ventricular arrhythmias [24]. This discrepancy may reflect survivorship bias: men, more likely to develop ventricular tachycardia and fibrillation (VT/VF) due to comorbidities such as insulin-treated diabetes and coronary artery disease (CAD), may experience higher out-of-hospital mortality and thus be underrepresented in hospitalized cohorts [25]. In contrast, women may be more susceptible to in-hospital CA due to delayed recognition of atypical symptoms or undiagnosed CAD [26,27,28].

Racial and socioeconomic disparities were also pronounced. Native American patients experienced a disproportionately large increase in CA incidence, while Black patients had steeper mortality increases compared to White patients. Individuals from lower socioeconomic strata showed both higher CA incidence and in-hospital mortality. These disparities likely stem from unequal access to guideline-directed therapies, underrecognized CAD, and inadequate metabolic risk management in historically marginalized populations [29,30]. Limited access to revascularization and atypical CAD presentations may further exacerbate poor outcomes. The persistently high in-hospital mortality in this cohort is consistent with previous studies and may, in part, be due to limited use of implantable cardioverter-defibrillators (ICDs). Although ICDs are primarily indicated for patients with HFrEF for the prevention of SCD, some patients with HFpEF may also experience lethal ventricular arrhythmias. However, because these patients do not meet traditional guideline-based criteria for primary prevention ICD placement, potential opportunities for life-saving intervention may be missed [31,32,33].

### 4.2. Shifting Comorbidity Profiles

The comorbidity landscape among elderly HFpEF patients experiencing in-hospital CA evolved significantly over the study period. While rates of diabetes without complications, uncomplicated hypertension, prior myocardial infarction (MI), and stroke declined, there was a marked rise in more complex, metabolically driven conditions such as diabetes with complications, complicated hypertension, hyperlipidemia, and obesity. These trends suggest a shift toward a more metabolically vulnerable HFpEF population, shaped by systemic inflammation, coronary microvascular dysfunction, and progressive myocardial fibrosis [34,35,36].

This evolving clinical profile parallels rising in-hospital mortality, reflecting broader stagnation in HFpEF outcomes amidst increasing comorbidity complexity [37]. Metabolic syndrome-related conditions, particularly complicated diabetes and hypertension may be key contributors to adverse prognoses. Obesity, linked to arrhythmogenic epicardial adipose tissue, further underlines the need for aggressive risk factor modification [38]. SGLT2 inhibitors are currently a standard of care in HFpEF following the results of EMPEROR-Preserved and DELIVER trials, which demonstrated significant reductions in heart failure hospitalizations across the spectrum of ejection fraction [37,39]. In addition to these cornerstone therapies, novel treatments targeting metabolic and inflammatory pathways are being explored. More recently, the STEP-HFpEF trial demonstrated that semaglutide, a GLP-1 receptor agonist, led to significant improvements in symptoms, physical function, and weight loss among patients with HFpEF and obesity, underscoring the potential of metabolic modulation in this population [40]. Additionally, finerenone, a non-steroidal mineralocorticoid receptor antagonist, was evaluated in the FINEARTS-HF trial, which reported a reduction in the composite of cardiovascular mortality and heart failure hospitalization [41]. Finerenone received FDA approval in July 2025 for treating patients with HFpEF and may soon be incorporated into the therapeutic landscape of HFpEF, particularly for patients with concurrent chronic kidney disease [42].

Interestingly, rates of prior MI declined despite CAD being a well-established contributor to HFpEF-related SCD [32]. This may reflect underdiagnosis, particularly among patients with atypical symptoms or those receiving less intensive diagnostic evaluation [29]. Given that up to two-thirds of HFpEF patients may harbor significant CAD (>50% stenosis), expanded CAD screening with coronary angiography may be a key opportunity to mitigate risk [43]. Unlike HFrEF, HFpEF lacks mortality-reducing guideline-directed therapies, and the contribution of arrhythmias to SCD in HFpEF remains poorly understood, necessitating further mechanistic studies [43,44].

### 4.3. Pathophysiological Insights

HFpEF is increasingly understood as a systemic disorder driven by comorbidities—particularly obesity, diabetes, and hypertension—that trigger endothelial dysfunction, nitric oxide (NO) deficiency, and cardiomyocyte remodeling. This pro-inflammatory state leads to coronary microvascular endothelial inflammation, reduced NO bioavailability, diminished protein kinase G (PKG) activity, and titin hypophosphorylation. These molecular alterations increase cardiomyocyte stiffness and interstitial fibrosis, resulting in elevated LV diastolic pressure and impaired relaxation [23,35,45,46,47,48]. Therapies such as statins, NO donors, and SGLT2 inhibitors have shown promise in targeting these mechanisms [35,37]. Equally critical is addressing disparities in care that may arise from underdiagnosed CAD, limited access to diagnostics such as coronary angiography, and persistent underrepresentation in clinical trials [38,49].

In older adults, age-related cardiovascular changes exacerbate these effects, further increasing vulnerability to CA [35,48,50]. Mitochondrial dysfunction and impaired myocardial energetics have also been implicated, highlighting the multifactorial nature of HFpEF [48]. These insights have led to the development of targeted therapies, including SGLT2 inhibitors, which may reduce oxidative stress, improve microvascular function, and enhance myocardial compliance [51,52]. This paradigm shift from a focus on afterload reduction to microvascular inflammation and metabolic modulation offers a more nuanced understanding of HFpEF and supports stratified treatment approaches [48,50,53].

### 4.4. Mortality and Healthcare Utilization

In-hospital mortality increased over the study period, with parallel trends across sexes. Black patients experienced the steepest mortality rise, and individuals from the lowest income quartile also showed significant increases. Despite a reduction in median hospital length of stay, average hospitalization charges rose, likely reflecting more intensive care without survival benefit. Nearly one-quarter of survivors were discharged to skilled nursing facilities, indicating substantial post-discharge disability and long-term care needs [54].

### 4.5. Limitations and Future Directions

This study relies on administrative data from the NIS, which depends on ICD-10-CM codes and may be subject to misclassification. We could not confirm clinical details such as ejection fraction, laboratory values, medications, or in-hospital interventions. We acknowledge that NIS data are limited by lack of rhythm-specific CA information, lack of sudden cardiac death data, and absence of details regarding underlying causes of CA. Second, the dataset includes only inpatient events, so it does not capture pre-hospital CAs or long-term outcomes. Third, residual confounding is possible, and the observational design prevents conclusions about causality. Furthermore, 2020 data may have been influenced by the COVID-19 pandemic, which could affect hospitalization patterns and case severity. Residual confounding is possible, and the observational design prevents conclusions about causality. Future work should link administrative and clinical data to improve phenotyping and follow patients over time. Studies exploring arrhythmic risk and targeted therapies in HFpEF are also needed, especially in diverse and high-risk populations.

## 5. Conclusions

Among elderly patients hospitalized with HFpEF, the incidence of in-hospital CA has increased over time, with persistently high mortality. Our analysis reveals significant disparities by race, sex, and socioeconomic status. Black, Hispanic, and low-income patients experienced the largest increases in CA and mortality. These trends suggest ongoing inequities in access to care, diagnostic evaluation, and management.

The evolving comorbidity profile, marked by a shift toward more complex metabolic disease, highlights the need for treatment approaches that address both cardiac function and systemic inflammation and metabolic risk. Although recent therapies such as SGLT2 inhibitors show promise, HFpEF remains without definitive mortality-reducing treatment. Improved inpatient risk stratification, targeted preventive strategies, and equitable access to guideline-based care will be essential to improve outcomes in this growing and vulnerable population.

## Figures and Tables

**Figure 1 jpm-15-00559-f001:**
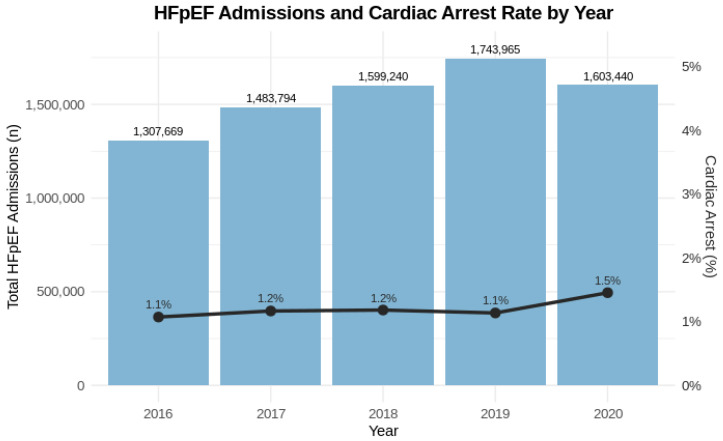
Trends in HFpEF Admissions and Cardiac Arrest Rates, 2016–2020. Bar graphs represent the total number of geriatric hospital admissions with heart failure with preserved ejection fraction (HFpEF) each year from 2016 to 2020. The overlaid line graph displays the corresponding annual percentage of admissions complicated by in-hospital cardiac arrest. While overall HFpEF admissions increased over time, the proportion of admissions associated with cardiac arrest remained relatively stable, with a slight increase noted in 2020. The primary y-axis indicates total admissions; the secondary y-axis reflects the cardiac arrest percentage. Values are annotated directly on the bars and line points for clarity.

**Table 1 jpm-15-00559-t001:** Trends in Cardiac Arrest Incidence and In-Hospital Mortality by Patient Demographics, Socioeconomic Status, and Hospital Characteristics (2016–2020). Cardiac arrest incidence and in-hospital mortality rates were evaluated across key demographic (age, sex, race/ethnicity), socioeconomic (income quartile), and hospital-level (region, rurality) variables. Cardiac arrest incidence was calculated as the percentage of patients experiencing cardiac arrest each year, while mortality rates reflect the proportion of patients with cardiac arrest who died during hospitalization. Significant temporal trends were observed across all subgroups (*p*-trend < 0.001), with exceptions noted in Appendix A.

Characteristic	Total *N* (%)	Cardiac Arrest Incidence, % (2016)	Cardiac Arrest Incidence, % (2020)	*p*-Trend	In-Hospital Mortality Among Arrest, % (2016)	In-Hospital Mortality Among Arrest, % (2020)	*p*-Trend
**Sex**							
Female	54%	1.0%	1.3%	<0.001	58.9%	62.3%	<0.001
Male	46%	1.3%	1.7%	<0.001	57.4%	61.1%	<0.001
**Race/Ethnicity**							
White	70%	0.9%	1.3%	<0.001	58.4%	60.8%	<0.001
Black	19%	1.7%	2.3%	<0.001	56.9%	62.9%	<0.001
Hispanic	8%	1.4%	2.0%	<0.001	62.6%	65.4%	0.510
Native American	<1%	1.0%	1.9%	<0.001	62.5%	64%	<0.001
**Median Household Income Quartile**							
Lowest (0–25%)	31%	1.2%	1.7%	<0.001	57.3%	63.5%	<0.001
Second (26–50%)	27%	1.0%	1.5%	<0.001	58.9%	61.2%	0.009
**Payer Type**							
Medicare	90%	1.0%	1.4%	<0.001	58.0%	61.5%	<0.001
**Hospital Region**							
Northeast	18%	0.87%	1.27%	<0.001	58.9%	64.2%	<0.001
**Rural vs. Urban**							
Rural	7%	0.77%	0.98%	<0.001	57.5%	33.7%	0.001
Urban	93%	1.11%	1.51%	<0.001	58.3%	61.4%	<0.001

**Table 2 jpm-15-00559-t002:** Selected Comorbidity Trends Among Elderly Patients with HFpEF and Cardiac Arrest, 2016 vs. 2020. This table highlights the prevalence of key cardiometabolic and vascular comorbidities in 2016 and 2020, along with *p*-values for temporal trends. Data reflect a shift toward more complex clinical profiles, including increased rates of diabetes with complications, obesity, and complicated hypertension, alongside a decline in conditions such as uncomplicated hypertension and diabetes without complications. All *p*-values were calculated using Pearson Chi-square tests.

Comorbidity	2016 (%)	2020 (%)	*p*-Trend
Diabetes without complications	17.8	7.3	<0.001
Diabetes with complications	28.0	42.9	<0.001
Hypertension (uncomplicated)	25.8	1.0	<0.001
Hypertension (complicated)	54.4	81.8	<0.001
Hyperlipidemia	46.5	51.0	<0.001
Obesity	21.8	25.2	<0.001
Prior myocardial infarction	10.6	8.3	<0.001
Prior stroke/TIA	9.4	8.7	0.118

**Table 3 jpm-15-00559-t003:** Trends in Hospital Length of Stay and Charges Among Elderly HFpEF Patients with Cardiac Arrest, 2016–2020. This table presents yearly trends in hospital length of stay (LOS) and hospitalization costs among elderly patients (≥65 years) admitted with HFpEF and cardiac arrest. LOS and charges are reported as medians with interquartile ranges (IQR). While the median LOS declined modestly from 7 to 6 days over the study period, median hospital charges increased by approximately 6%, reflecting rising healthcare costs. *p*-values for trend were statistically significant (*p* < 0.001) as determined by Kruskal–Wallis tests.

Year	Median LOS (Days)	IQR (Days)	Median Hospital Charges (USD)	IQR Charges (USD)
2016	7	3–13	$105,461	$47,448–$224,968
2017	7	3–13	$107,302	$46,810–$224,095
2018	6	3–13	$105,431	$50,001–$212,849
2019	6	3–13	$107,084	$48,103–$227,540
2020	6	3–13	$112,447	$50,141–$233,185
Overall	6	3–13	$107,588	$48,780–$224,968

## Data Availability

The data that support the findings of this study are publicly available from the Healthcare Cost and Utilization Project (HCUP) at [https://www.hcup-us.ahrq.gov/].

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
