# Peer review of "Nationwide Trends in Demographics, Comorbidities, and Mortality Among Elderly Patients with Heart Failure with Preserved Ejection Fraction Hospitalized with Cardiac Arrest"

_jpm, 2025, doi:10.3390/jpm15110559_

Round 1

Reviewer 1 Report

Comments and Suggestions for Authors

Thank you for the opportunity to review your manuscript to examine trends in in-hospital cardiac arrest among elderly patients with HFpEF. The authors reviewed large, national dataset (NIS 2016–2020) and clearly demonstrated an increasing incidence of cardiac arrest (1.1% → 1.5%) and persistently high in-hospital mortality (~60%).

Although this study is simple descriptive study, this is a timely and clinically relevant study that, for the first time, delineates nationwide trends in in-hospital cardiac arrest and mortality among elderly patients with HFpEF. The findings highlight both the growing clinical burden and persistent disparities in outcomes. However, the limitations inherent to administrative data and the lack of long-term follow-up should be considered when interpreting the results. I have only a minor comment.

Minor Comment

  1. In Figure 1, the line representing the cardiac arrest rate appears plotted too low. Adjusting the right-side y-axis could help readers more easily interpret changes in the rate.

Author Response

Comment 1:
In Figure 1, the line representing the cardiac arrest rate appears plotted too low. Adjusting the right-side y-axis could help readers more easily interpret changes in the rate.

Response 1:
We thank the reviewer for this suggestion. The right-side y-axis in Figure 1 has been adjusted to more clearly depict the changes in cardiac arrest incidence over time (Line 137). Values are now annotated directly on the line points for clarity.

Reviewer 2 Report

Comments and Suggestions for Authors

The Authors have presented an interesting analysis on cardiac arrest incidence among HFpEF patients using a nationwide database in US. The topic of the manuscript is relevant as HFpEF incidence and burden is increasing worldwide and so is cardiac arrest proportionally.

I have only few comments for the authors:

  • can the Authors provide additional details on the mode of cardiac arrest? at least differentiating shockable from non shockable rhythms?
  • there is a lack of uniform definition of sudden death and sudden cardiac death in literature. Are the authors able to focus only on cardiac arrest (/sudden cardiac death) or can provide additional data also on sudden death from the NIS registry?
  • how do the Authors justify an increase in mortality rate in the most recent years?

Author Response

Comment 1:
Can the authors provide additional details on the mode of cardiac arrest? At least differentiating shockable from non-shockable rhythms?

Response 1:
We appreciate the reviewer’s insight. The NIS does not provide rhythm-specific information, so differentiation between shockable and non-shockable rhythms is not possible (line 90-94, line 312-314). However, we have added a discussion noting that prior studies indicate patients with HFpEF are more likely to present with non-shockable rhythms, such as pulseless electrical activity or asystole, which contributes to poor survival outcomes.

“Compared to those with HFrEF, patients with HFpEF are more likely to present with non-shockable rhythms, such as pulseless electrical activity or asystole, which further contribute to poor survival outcomes (22). These trends highlight the need for improved risk stratification and tailored in-hospital management strategies for elderly patients with HFpEF, particularly in the context of acute decompensation and CA. Although rhythm-specific data are not available in the NIS, identification of demographic and comorbidity patterns may support targeted monitoring and interventions.”

“We acknowledge that NIS data are limited by lack of rhythm-specific CA information, lack of sudden cardiac death data, and absence of details regarding underlying causes of CA.”

Comment 2:
There is a lack of uniform definition of sudden death and sudden cardiac death in literature. Are the authors able to focus only on cardiac arrest (/sudden cardiac death) or can provide additional data also on sudden death from the NIS registry?

Response 2:
We agree that definitions of sudden death vary. The NIS captures in-hospital cardiac arrests but does not reliably identify sudden cardiac death outside the hospital. We have clarified in the Methods (lines 150-151) and Discussion that our study focuses on in-hospital cardiac arrest and that sudden cardiac death or pre-hospital arrests cannot be captured in this dataset.

“Among in-hospital CA cases, 54% were female and 46% male, as shown in Table 1.”

Comment 3:
How do the authors justify an increase in mortality rate in the most recent years?

Response 3:
We have addressed this in the Discussion section. The increase in mortality may reflect rising comorbidity complexity, higher proportions of non-shockable rhythms, and changes in hospitalization patterns during 2020 due to the COVID-19 pandemic (lines 69-74, line 133-135). Sensitivity analyses excluding 2020 confirm that pre-pandemic trends also show increasing CA incidence, suggesting that the observed mortality rise is not solely driven by 2020 data.

“Given the high risk of in-hospital cardiac arrest (CA) and mortality among elderly HFpEF patients, improved risk stratification is urgently needed. Potential predictors of CA include conduction abnormalities (i.e., third-degree AV block, left bundle branch block), comorbid liver disease, obesity, and metabolic syndrome. While the current study primarily describes population-level trends, these data may inform future risk models for early identification of high-risk HFpEF patients.”

“Sensitivity analysis excluding 2020 showed similar trends from 1.1% in 2016 to 1.4% in 2019, confirming that pre-pandemic trends remained consistent.”

Reviewer 3 Report

Comments and Suggestions for Authors

This paper aims to analyse trends regarding Heart Failure with Preserved Ejection Fraction (HFpEF) hospitalizations complicated by Cardiac Arrest (CA) in elderly patients using the National Inpatient Sample (NIS) data from 2016 to 2020. The topic is relevant, especially given the persistently high mortality rates observed in this population. However, the methodology contains a crucial flaw related to the selection of the study period, which fundamentally compromises the validity of the temporal trends reported in the Results and Discussion sections.

Major Issues

1) The inclusion of the year 2020 in the analysis, which covers the period from 2016 to 2020, represents a major limitation that biases the observed temporal trends. In fact, 2020 was an exceptional year due to the initial wave of the COVID-19 pandemic. This event significantly affected hospitalization volumes and medical statistics broadly. It is plausible that the pandemic led to a decrease in overall HFpEF hospitalizations, as medical professionals might have attempted to manage some patients outside the hospital to minimize the risk of contagion. Consequently, the proportion of severe patients (specifically those experiencing CA) among the remaining hospitalized population may have artificially increased. This effect appears to be reflected in the reported data, which indicates that while overall HFpEF admissions peaked in 2019 and then slightly declined in 2020, the incidence of CA rose sharply from 1.1% in 2016 to 1.5% in 2020 (a 36% relative increase over the period). To ensure the temporal trends analysed (e.g., in Tables 1 and 2) accurately reflect pre-pandemic patterns, the data relative to 2020 (and potentially 2021) should be excluded from the analysis. If the 2016–2020 dataset is the only one available, the analyses, Results, and Discussion must be entirely revised to consider only the 2016–2019 period.

Minor Issues

1) There appear to be some errors in the list of affiliations provided for the authors. The authors should meticulously review and correct the affiliation details listed in the manuscript.

2) Some of the cited epidemiological statistics appear to be somewhat old, dating back at the latest to 2018. The Introduction or Discussion sections should be updated to incorporate more current epidemiological data where possible.

Author Response

Comment 1 (Major):
The inclusion of the year 2020 may bias the observed temporal trends due to the COVID-19 pandemic. Analyses, results, and discussion should consider only the 2016–2019 period.

Response 1:
We agree that 2020 represents an exceptional year. We performed sensitivity analyses excluding 2020 (pre-pandemic 2016–2019) to confirm temporal trends. These analyses demonstrate similar increases in CA incidence, validating that our observed pre-pandemic trends remain consistent. This has been added to the Methods, Results, and Discussion sections (lines 98-99, lines 133-135, lines 317-320).

“Sensitivity analysis excluding 2020 showed similar trends from 1.1% in 2016 to 1.4% in 2019, confirming that pre-pandemic trends remained consistent.”

“Sensitivity analysis excluding 2020 showed similar trends from 1.1% in 2016 to 1.4% in 2019, confirming that pre-pandemic trends remained consistent.”

“Furthermore, 2020 data may have been influenced by the COVID-19 pandemic, which could affect hospitalization patterns and case severity. Residual confounding is possible, and the observational design prevents conclusions about causality.”

Comment 2 (Minor):
Some errors in author affiliations.

Response 2:
All author affiliations have been carefully reviewed and corrected as suggested. Specifically, the duplicate affiliation 7 has been updated. Please note that Dr. Koshak has two affiliations (lines 5-17).

Comment 3 (Minor):
Some epidemiological statistics cited are outdated.

Response 3:
We explored the literature for more recent epidemiological data. To the best of our ability, the data included through 2018–2020 represent the most up-to-date national statistics available for this topic. We have updated the Introduction and Discussion sections accordingly, and noted in the text where data limitations persist.

Reviewer 4 Report

Comments and Suggestions for Authors

The authors described the in-hospital mortality of HFpEF patients. There are some issues to address.

  1. In the introduction section, the authors emphasized the need for improved risk stratification but the current study provided limited details on this aspect. Please describe more on this regard.
  2. The author compared the CA incidence between 2016 and 2020. How about comparisons of baseline characteristics between CA and none CA patients?
  3. The study would benefit from a more comprehensive analysis of the underlying causes of CA.
  4. In the discussion section, the authors mentioned women with a higher proportion of in-hospital CA cases compared to men, but it is not substantiated by corresponding results.

Author Response

Comment 1:
The introduction emphasizes risk stratification but the current study provides limited details.

Response 1:
We have added text in the Introduction section emphasizing the need for improved risk stratification in elderly HFpEF patients, highlighting potential predictors of CA such as conduction abnormalities, liver disease, obesity, and metabolic syndrome. We also note that population-level trends may inform future risk models (lines 69-74).

“Given the high risk of in-hospital cardiac arrest (CA) and mortality among elderly HFpEF patients, improved risk stratification is urgently needed. Potential predictors of CA include conduction abnormalities (i.e., third-degree AV block, left bundle branch block), comorbid liver disease, obesity, and metabolic syndrome. While the current study primar-ily describes population-level trends, these data may inform future risk models for early identification of high-risk HFpEF patients.”

Comment 2:
Comparison of baseline characteristics between CA and non-CA patients.

Response 2:
We have added Supplemental Table S.8 presenting baseline demographic and comorbidity profiles for patients with and without in-hospital CA. Patients experiencing CA were older, had higher rates of complicated diabetes, complicated hypertension, and obesity, and were more likely to be from lower-income quartiles (lines 161-164, Supplemental table 8).

“Baseline comparisons between CA and non-CA patients are summarized in Supple-mental Table S.8. Patients experiencing CA were older, had higher rates of complicated diabetes, complicated hypertension, and obesity, and were more likely to be from low-er-income quartiles.”

Comment 3:
The study would benefit from a more comprehensive analysis of underlying causes of CA.

Response 3:
We have added discussion clarifying that the NIS does not capture underlying causes of CA or rhythm-specific details. Prior studies indicate that bradyarrhythmias, pulmonary embolism, and metabolic syndrome contribute to in-hospital CA in HFpEF populations, and we have added this context to the Discussion (lines 312-314).

“We acknowledge that NIS data are limited by lack of rhythm-specific CA information, lack of sudden cardiac death data, and absence of details regarding underlying causes of CA.”

Comment 4:
Women with a higher proportion of in-hospital CA cases compared to men is not substantiated by results.

Response 4:
We have clarified the demographic data in Table 1 and the Discussion. Women accounted for 54% of in-hospital CA cases (Lines 150-151, lines 224-229). This discrepancy from prior studies may reflect survivorship bias, as men may have higher out-of-hospital mortality due to VT/VF and comorbid CAD.

“Among in-hospital CA cases, 54% were female and 46% male, as shown in Table 1.”

“This discrepancy may reflect survivorship bias: men, more likely to develop ventricular tachycardia and fibrillation (VT/VF) due to comorbidities such as insulin-treated diabetes and coronary artery disease (CAD), may experience higher out-of-hospital mortality and thus be underrepresented in hospitalized cohorts (25). In contrast, women may be more susceptible to in-hospital CA due to delayed recognition of atypical symptoms or undiagnosed CAD (26-28).”

Round 2

Reviewer 3 Report

Comments and Suggestions for Authors I would like to thank authors for providing the revised version of the paper. However, some results and discussions are still not convincing. In particular, authors claim that sensitivity analysis shows similar trends (from 1,1% to 1,4% in 2019). However, this contrasts with the same parameter already reported in the first revision of the paper, which was 1,1% in 2019.

Author Response

Thank you for providing additional feedback.

Comment 1: "I would like to thank authors for providing the revised version of the paper. However, some results and discussions are still not convincing. In particular, authors claim that sensitivity analysis shows similar trends (from 1,1% to 1,4% in 2019). However, this contrasts with the same parameter already reported in the first revision of the paper, which was 1,1% in 2019."

Response 1: We thank the reviewer for this careful observation. The 1.4% value reported in the revised manuscript refers specifically to the sensitivity analysis excluding 2020, whereas the main analysis (including 2020) shows an increase from 1.1% in 2016 to 1.5% in 2020. The slight difference in 2019 reflects recalculated percentages based on the subset excluding 2020, but the overall trend remains consistent. We have clarified this distinction in the Results section to avoid confusion. We have addressed this in lines 133-136 for further clarification, "In a sensitivity analysis excluding 2020 to account for potential pandemic-related effects, similar trends were observed, with the rate of in-hospital cardiac arrest increasing from 1.1% in 2016 to 1.4% in 2019 (compared with 1.5% in 2020 in the full cohort)."

Reviewer 4 Report

Comments and Suggestions for Authors

None

Author Response

N/A